# Self-Reported Long COVID in the General Population: Sociodemographic and Health Correlates in a Cross-National Sample

**DOI:** 10.3390/life12060901

**Published:** 2022-06-15

**Authors:** Tore Bonsaksen, Janni Leung, Daicia Price, Mary Ruffolo, Gary Lamph, Isaac Kabelenga, Hilde Thygesen, Amy Østertun Geirdal

**Affiliations:** 1Department of Health and Nursing Science, Inland Norway University of Applied Sciences, 2418 Elverum, Norway; 2Department of Health, VID Specialized University, 4024 Stavanger, Norway; 3Faculty of Health and Behavioural Science, The University of Queensland, Brisbane, QLD 4072, Australia; j.leung1@uq.edu.au; 4School of Social Work, University of Michigan, Ann Arbor, MI 48109, USA; daiciars@umich.edu (D.P.); mruffolo@umich.edu (M.R.); 5School of Nursing, The University of Central Lancashire, Preston PR1 2HE, UK; glamph@uclan.ac.uk; 6Department of Social Work and Sociology, School of Humanities and Social Sciences, University of Zambia, Lusaka P.O. Box 32379, Zambia; isaackabelenga@gmail.com; 7Department of Occupational Therapy, Prosthetics and Orthotics, Faculty of Health Sciences, Oslo Metropolitan University, 0176 Oslo, Norway; hilde.thygesen@oslomet.no; 8Department of Health, Faculty of Health Studies, VID Specialized University, 0370 Oslo, Norway; 9Department of Social Work, Faculty of Social Sciences, Oslo Metropolitan University, 0176 Oslo, Norway; amyoge@oslomet.no

**Keywords:** COVID-19, cross-cultural study, fatigue, mental health, stress, psychological distress

## Abstract

We aimed to gain knowledge of possible sociodemographic predictors of long COVID and whether long COVID was associated with health outcomes almost two years after the pandemic outbreak. There were 1649 adults who participated in the study by completing a cross-sectional online survey disseminated openly in Norway, the UK, the USA, and Australia between November 2021 and January 2022. Participants were defined as having long COVID based on self-reports that they had been infected by COVID-19 and were experiencing long-lasting COVID symptoms. Logistic regression analyses were used to examine possible sociodemographic predictors, and multivariate analysis of variance was used to examine whether long COVID status was associated with health outcomes. None of the sociodemographic variables was significantly associated with reporting long COVID. Having long COVID was associated with higher levels of psychological distress, fatigue, and perceived stress. The effect of long COVID on health outcomes was greater among men than among women. In conclusion, long COVID appeared across sociodemographic groups. People with long COVID reported worsened health outcomes compared to those who had had COVID-19 but without long-term symptoms. Men experiencing long COVID appear to be particularly vulnerable to experiencing poorer health outcomes; health services may pay extra attention to potentially unnoticed needs for support among men experiencing long COVID.

## 1. Introduction

Two years after the world was cast into the coronavirus (SARS-CoV-2) pandemic, more than 445 million people globally have been infected and almost six million have died from the coronavirus disease 2019 (COVID-19) [1]. COVID-19 is a respiratory infection that has been seen to spread quickly through contact with others. Among those infected, illness severity has varied widely between individuals and populations [2]. While some have died and some have become seriously ill, many have experienced mild to moderate symptoms, such as fever, cough, and sore throat [3,4], similar to symptoms of seasonal influenza. An additional common symptom has been the loss of taste and smell [4]. However, others infected by COVID-19 have been practically asymptomatic.

For most people infected with COVID-19, the illness has had a relatively short duration. However, as people who have experienced COVID-19 have shared their experiences, awareness that the effects of COVID-19 may extend beyond the initial infection has increased. A study from the USA found that 65% (175 of 270) had returned to usual health seven days after a positive COVID-19 test, while 35% (95 of 270) had not returned to usual health when interviewed 2–3 weeks after testing positive [5]. The risk of not having returned to usual health at the time of the interview was higher for people of older age and people with chronic illnesses, psychiatric conditions, and obesity [5]. These results correspond with studies of risk factors for severe COVID-19 illness, suggesting more severe illness among people who are older, obese, and with higher levels of comorbidity [2,6,7]. Studies have also found higher risk of severe illness among males [2,6,7] and among people with Asian [2] and African-American backgrounds [6]. While genetic, hormonal, and immunological gender differences may contribute to explain the higher risk of severe COVID-19 illness in men [8,9], social mechanisms linked to poverty, deprivation, and social exclusion are salient explanations for ethnicity-based differences in illness severity [2].

Long COVID refers to symptoms lasting relatively long after the initial viral infection. Diagnostic guidelines differentiate between post-acute COVID-19, defined by signs and symptoms persisting 4–12 weeks after illness onset, and post-COVID-19 syndrome, defined by signs and symptoms with a duration of more than 12 weeks after illness onset [10]. Long COVID encompasses both conditions, thus denoting COVID-related symptoms lasting longer than four weeks after becoming ill. Studies have pointed towards a high prevalence of long COVID among people who have been infected with COVID-19 [4,11,12,13,14], and female gender has been found to be the most prominent risk factor for experiencing long COVID in adults [13] as well as in children and adolescents [15]. In England, Ayoubkhani and coworkers [12] found that nearly one in three patients previously hospitalized with acute COVID-19 were readmitted to hospital over a mean follow-up period of 140 days. In Russia, Munblit and coworkers [13] examined patients discharged from four hospitals in Moscow, and found that symptoms persisted over a median of 218 days post-discharge for 47% of the patients. In a recent systematic review, more than 50% of COVID-19 patients were found to experience at least one symptom at follow-ups of up to 6 months [14].

Symptoms of long COVID are often similar to those experienced in the acute phase of the illness. According to Groff and colleagues [14], symptoms may be categorized as neurological (e.g., headaches, memory deficits, concentration problems), mental health-related (e.g., depression, anxiety, post-traumatic stress, sleep disturbance), pulmonary (e.g., dyspnea, cough), mobility-related (e.g., general functioning, reduced exercise tolerance), general or constitutional (e.g., fatigue, muscle weakness, general pain), cardiovascular (e.g., chest pain, palpitation), gastrointestinal, dermatological, and ear, nose, and throat-related (e.g., diarrhea, vomiting, sore throat). In particular, fatigue, shortness of breath, and sleep problems have been found to be commonly experienced [11,16,17], and levels of fatigue and several other symptoms appear to be particularly high among young women [11]. According to a systematic review of studies on long-term effects of COVID-19 [14], fatigue or muscle weakness (median: 37.5%), generalized anxiety disorder (median: 29.6%), and difficulties with concentration (median: 23.8%) are among the most frequently occurring symptoms. Two main lines of interpretation of the long COVID condition have been described. It may be understood as a direct long-term effect of the viral infection, possibly explained by persistent viremia, relapse or reinfection, hyperinflammatory immune response, autoimmunity, or cytokine- or hypoxia-induced injury. Alternatively, long COVID may be interpreted as indirect effects on mental health due to increased stress, social isolation, and economic problems instigated by the pandemic, such as loss of employment [14]. Being without employment and experiencing isolation during the pandemic have been associated with poorer mental health outcomes [18,19,20,21]. Possibly, long COVID symptoms may also be related to an individual’s access to resources, such as knowledge (as indicated by education level) and a close partner who can provide support. 

While research is ongoing and growing, current knowledge about long COVID, its risk factors and potential consequences, is still relatively sparse. Consequently, the mechanisms underpinning the long COVID experience are not well understood. More research is needed in order to better predict which groups are more inclined to experience long-lasting symptoms and to target these groups in healthcare interventions. More research is also needed to gain a comprehensive overview of the probable outcomes related to long COVID.

### Aim of the Study

The aim of the study was to gain knowledge about the factors associated with long COVID, both sociodemographic and health-related. Specifically, we examined (i) whether sociodemographic factors were associated with long COVID status; (ii) whether long COVID status was associated with three health outcomes: psychological distress, fatigue, and perceived stress; and (iii) whether gender moderated the associations between long COVID status and health outcomes. 

## 2. Materials and Methods

### 2.1. Design

The study reports from the third cross-sectional survey disseminated openly in four countries (Norway, the United Kingdom (UK), the USA, and Australia) during the COVID-19 pandemic. This survey was open for the general public’s participation between November 2021 and January 2022, while the two previous surveys were administered in April/May 2020 and in November 2020, respectively.

### 2.2. Sample 

The sample comprised 1,649 participants. Among them, 242 (14.7%) were from Norway, 255 (15.5%) were from the UK, 915 (55.5%) were from the USA, and 237 (14.4%) were from Australia. In the total sample, 310 (18.8%) reported to have had COVID-19 infection. Seven of the 310 did not report on long COVID status. Of the remaining 303 participants, 87 (28.7%) reported having long COVID. An overview of all participants, participants with COVID-19, and participants with long COVID is provided in Table 1.

### 2.3. Measures

#### 2.3.1. COVID-19 Infection and Self-Reported Long COVID

One question was used to examine whether the participants had experienced COVID-19 infection: “Have you been infected by COVID-19?” The response options were ‘yes’ and ‘no’. Those affirming that they had been infected were also given a follow-up question about long COVID: “Are you experiencing long-lasting or chronic COVID symptoms?” Response options were ‘yes’ and ‘no’. Those who responded ‘yes’ to the question about long COVID were classified as ‘long COVID’, while those who responded ‘no’ were classified as ‘not long COVID’. In the following, the term ‘long COVID’ is used when referring to ‘self-reported long COVID’.

#### 2.3.2. Sociodemographic Characteristics

Sociodemographic variables included age group (18–29, 30–39, 40–49, 50–59, 60–69, 70 and above), gender (male vs. female), education level (lower vs. bachelor’s degree or higher), being in employment (yes/no), and having a spouse/partner (yes/no).

#### 2.3.3. Psychological Distress

The 12-item General Health Questionnaire (GHQ-12) was used to measure psychological distress [22,23]. Its validity across samples and contexts has been demonstrated in a large number of studies in general adult, clinical, work, and student populations [23,24,25,26,27], and translations have been made in several languages, including Norwegian [28]. Six items of the GHQ-12 are phrased positively (e.g., ‘able to enjoy day-to-day activities’), while six items are phrased negatively (e.g., ‘felt constantly under strain’). For each item, the person indicates the degree to which he or she has experienced the item content during the past two weeks (‘less than usual’, ‘as usual’, ‘more than usual’, or ‘much more than usual’). Items are scored between 0 and 3, and positively formulated items are recoded prior to analysis. As a result, the GHQ-12 scale score range is 0–36, with higher scores indicating higher levels of psychological distress. In the current sample, Cronbach’s α was 0.91.

#### 2.3.4. Fatigue

The 14-item Chalder Fatigue Scale was used to measure fatigue [29]. Example items include “Recently, are you lacking in energy?” and “Recently, do you have difficulty concentrating?”. All items are rated on a 4-point Likert scale (0 = better than usual, 1 = no more than usual, 2 = worse than usual, 3 = much worse than usual), with higher scores indicating greater fatigue. The scale has been used in several studies and translated into several languages. It has often been used with the initially proposed two-factor solution, separating physical and mental aspects of fatigue, or even with three or four factors [30,31,32]. However, an overall measure of fatigue was required in this study, so we constructed each participant’s fatigue score as the sum of all the item scores, in line with Chilcot and co-workers [33]. In the current sample, Cronbach’s α was 0.93.

#### 2.3.5. Perceived Stress

The 10-item Perceived Stress Scale (PSS-10) [34,35] has been used in several community studies in different countries and translated into several languages [36]. It is one of the most widely disseminated self-report scales that provides a global stress score based on general questions. It measures the degree to which an individual appraises their life situations as stressful. The questions capture how out of control or overloaded an individual has felt. The responses use a Likert scale (4 = very often, 3 = fairly often, 2 = sometimes, 1 = almost never, and 0 = never). Each question starts with, “In the last month …”, and is followed up with (example items) “… how often have you been upset because of something that happened unexpectedly?” and “… how often have you felt that you were unable to control the important things in your life?” The higher the score on this scale, the greater the level of perceived stress. Previous studies have shown high (0.83) Cronbach’s α values for the PSS-10 [37]. In the current sample, Cronbach’s α was 0.90.

### 2.4. Statistical Analysis

We conducted our analysis by comparing the long COVID and not long COVID (the sample who had COVID-19 but without long COVID) groups. Comparisons of proportions between groups were made using the chi-square test. Single and multiple logistic regression analyses were used to examine factors associated with long COVID status. Independent variables were age group, gender, education level, having a spouse or partner, and employment. Age group was used as a continuous variable, while all other independent variables were binary categorical variables. Odds ratios (ORs) was used as indices of effect size, and the 95% confidence intervals (95% CI) of the ORs were reported. Health outcomes associated with long COVID were examined with multivariate analysis of variance (MANOVA). Three dependent variables were included: summarized ratings on the GHQ-12, the Fatigue scale, and the PSS-10. Long COVID status and gender were used as fixed factors, and we examined the interaction effect of long COVID × gender to assess whether the effects of long COVID on health outcomes differed between men and women. Age group was included as a covariate. If the multivariate test of an effect of long COVID status on the dependent measures was statistically significant, we would proceed to examine the specific effects on each of the health outcomes. In pairwise comparisons, Bonferroni correction was applied to the significance levels to adjust for inflating error levels. Effect sizes were reported as partial *η*^2^, and statistical significance was set at *p* < 0.05.

### 2.5. Ethics

The study was conducted after receiving ethical approval from the following review boards: OsloMet (20/03676) and the regional committees for medical and health research ethics (REK; ref. 132066) in Norway; the University of Michigan Institutional Review Board for Health Sciences and Behavioral Sciences (IRB HSBS), which designated the study as exempt (HUM00180296) in the USA; the University of Central Lancashire (Health Ethics Review Panel) (HEALTH 0246) in the UK; and the University of Queensland Human Research Ethics Committees in Australia (HSR1920-080 2020000956).

## 3. Results

### 3.1. Long COVID in Sample Subgroups

Overall, 303 (18.4%) individuals in our sample reported COVID-19 infection and, among these, 87 (28.7%) reported long COVID. COVID-19 reports were higher in the UK and USA, and lower in Norway and Australia (see Table 1). Among those with COVID-19, more individuals in our UK sample reported long COVID compared to the USA sample. 

Table 2 displays the proportions with long COVID by sociodemographic factors, with the results of significance tests for differences. No tests were statistically significant, indicating that long COVID cases did not significantly differ between sample subgroups. 

### 3.2. Associations between Sociodemographic Factors and Long COVID Status

Table 3 displays the results from the single and multiple logistic regression analyses. The analyses confirmed the initial results, indicating no significant associations between any of the sociodemographic variables and long COVID status. The change in ORs from the unadjusted to the adjusted analyses were negligible, indicating no suppressor effects. The regression model was not statistically significant (*p* = 0.35) and estimates of explained variance were very small (R^2^ ranging between 1.9% and 2.8%), indicating that the sociodemographic variables were largely unable to explain variations in long COVID status.

### 3.3. Effect of Long COVID on Health Outcomes

Box’s test of equality of covariance matrices was not statistically significant, indicating that the observed covariance of the dependent variables (ratings on the psychological distress, fatigue, and perceived stress scales) was equal across groups. Levene’s tests of equality of error variances were non-significant for GHQ-12 and PSS-10, but significant (*p* = 0.01) for the fatigue scale, indicating a pattern of residuals deviating from the normal distribution on this scale. 

The multivariate tests revealed that long COVID status was significantly associated with the three health outcomes (Wilk’s lambda = 0.87, *F* (3279) = 14.3, *p* < 0.001, partial *η*^2^ = 0.13). The outcomes were also significantly associated with age (Wilk’s lambda = 0.95, *F* (3279) = 4.57, *p* < 0.01, partial *η*^2^ = 0.05), gender (Wilk’s lambda = 0.97, *F* (3279) = 2.97, *p* < 0.05, partial *η*^2^ = 0.03), and the interaction between long COVID status and gender (Wilk’s lambda = 0.97, *F* (3279) = 3.21, *p* < 0.05, partial *η*^2^ = 0.03). 

The effects of the independent variables on each of the outcomes are displayed in Table 4. Long COVID was significantly associated with ratings on the psychological distress, fatigue, and perceived stress scales. Age was significantly associated with ratings on the GHQ-12 and PSS-10 scales, whereas the association between age and ratings on the fatigue scale did not reach statistical significance. Post hoc one-way ANOVA analyses revealed that ratings on the psychological distress, fatigue, and perceived stress scales were consistently lower in the higher age groups (data not shown). Gender was not associated with any of the individual outcomes. Gender significantly moderated the effect of long COVID on psychological distress and fatigue, but not on perceived stress.

Estimated marginal means for each of the outcome measures by group, adjusted for age, are displayed in Table 5. Participants with long COVID had significantly higher levels of psychological distress, fatigue, and perceived stress, with effect sizes ranging between 0.03 (PSS-10) and 0.13 (fatigue), compared to participants without long COVID. While gender had no effect on any of the outcomes, gender significantly moderated the effect of long COVID on psychological distress and fatigue. While outcome ratings among participants with long COVID were poorer for both genders, the effect of long COVID was stronger among men than among women. Among men, the estimated means were considerably higher for GHQ-12 (M = 19.4) and fatigue (M = 25.1) among participants with long COVID compared to the corresponding estimated means among those without long COVID (M = 12.9 and M = 16.7, respectively). Among women, the differences in estimated means between those with and without long COVID were smaller. The moderation effects of gender are illustrated in Figure 1 (psychological distress), Figure 2 (fatigue), and Figure 3 (perceived stress).

## 4. Discussion

### 4.1. Summary of Findings

This study showed that 29% of those who had experienced COVID-19 also experienced longstanding or chronic COVID-19 symptoms. None of the included sociodemographic variables were significantly associated with long COVID. Participants with long COVID had significantly higher levels of psychological distress, fatigue, and perceived stress compared to participants who had experienced COVID-19 but were no longer symptomatic. Gender moderated the association between long COVID status, and psychological distress and fatigue. While participants with long COVID generally perceived more psychological distress, fatigue, and perceived stress than those without long COVID, differences in two of the three outcomes were larger for men than for women.

### 4.2. Prevalence of Long COVID

This study showed that 29% of participants who had experienced COVID-19 also reported longstanding or chronic COVID-19 symptoms. As all information in this study is based on self-reports, there is substantial uncertainty associated with this prevalence measure. Self-report measures tend to yield a higher frequency of cases compared to the frequencies obtained by clinical diagnosis [38]. However, at face value, the prevalence found in this study is similar to that reported by Tenforde and colleagues [5], but lower than the prevalence rates (about 50%) reported in other studies [13,14]. In general, prevalence rates of long COVID are difficult to interpret in the absence of commonly and rigidly applied diagnostic criteria and in the context of widely varying research methodologies.

### 4.3. Sociodemographic Variables Associated with Long COVID

The study showed that none of the sociodemographic variables was significantly associated with long COVID. While previous research has found that the prevalence of long COVID is higher among women than among men, indicating that female gender is a sociodemographic risk factor for long COVID in adults [13] and in children and adolescents [15], this finding was not substantiated in our study. Moreover, our study cannot point in the direction of other possible risk factors among the sociodemographic variables. 

### 4.4. Health Outcomes Associated with Long COVID 

As expected, participants with long COVID reported higher levels of psychological distress, fatigue, and perceived stress than participants who had experienced COVID-19 infection but without the long-term symptoms. Following the views of Groff and co-workers [14], these symptoms may be understood as direct long-term effects of COVID-19 infection, with a starting point in the physiological response, or they may reflect responses to a combination of stressors experienced during the pandemic situation. Considering the vast amount of evidence related to the pandemic’s negative mental health effects in the general population [39,40,41], including the burden of isolation and loneliness [42], long COVID may not only be a consequence of infection; it may also be a response to loneliness, uncertainty, and stress experienced during the pandemic.

There was no direct gender effect on health outcomes, indicating that men and women who had experienced COVID-19 had similar levels of psychological distress, fatigue, and perceived stress. However, gender moderated the effect of long COVID on psychological distress and fatigue, implying that the differences in psychological distress and fatigue between men with and without long COVID were larger than the corresponding differences between women with and without long COVID. Thus, while women are more inclined to get long COVID, as reported in the literature [13,15], men who do get long COVID appear to experience more severe symptoms. Possibly, the male participants may have been more severely ill in the acute phase. This would be in line with research results indicating a higher risk of severe COVID-19 illness among males [2,6,7], such that more severe symptoms during the post-acute phase may be reflective of the initial symptomatic burden.

A second possible explanation concerns differences in reporting patterns in men and women, rather than ‘real’ differences in symptom levels. Considering the evidence related to influenza, Sue [43] provocatively asked whether “men are wimps or just immunologically inferior”. Much evidence supports a notion of men having higher morbidity and mortality in response to influenza compared to women [44,45]. While the reasons for this are not fully understood, there may be different immune responses, including different responses to vaccines, between men and women [8,9,43]. Hormonal constitution may be partly responsible for this gender difference, as an immunorepressive effect of testosterone has been suggested [46]. Thus, applying the argument to long COVID, men with long COVID may indeed experience substantially poorer health than their male peers without long COVID, while the difference may be smaller between women with and without long COVID. Interestingly, and despite men’s lower inclination to report and seek help for mental health problems [47], men’s perception of poorer health when having long COVID appears to include higher levels of psychological distress and fatigue. 

### 4.5. Study Limitations

Data for the study were collected by means of an online survey with participants recruited by self-selection. Descriptively, women and younger people were overrepresented, whereas older people over the age of 70 were particularly underrepresented. Thus, the sample is not representative of the general population in the countries included. The data are cross-sectional, and we are unable to infer causation. Those with poorer health at the start of the pandemic could have been more likely to contract long COVID. However, we do not have access to data about the participants other than those that were reported in the survey. The participants reporting having had COVID-19 were asked whether or not they experienced long-lasting or chronic COVID symptoms, and the responses to this question were used to distinguish between long COVID cases and non-cases. It is possible that the result of this classification procedure was not fully consistent with the established definition of long COVID, which requires that symptoms last more than four weeks after illness onset [10]. In turn, incorrect classifications may have skewed the results. The sample sizes were small in some socio-demographic subgroups, including the group of males with long COVID. As a result, low statistical power may account for some of the non-significant results. While we cannot conclude that long COVID does not differ by sociodemographic variables, we observed that long COVID was reported by a substantial proportion of people across sociodemographic groups. The results related to the fatigue scale should be treated cautiously, as the distribution of residuals deviated from the normal distribution. Thus, the scale was not ideal for inclusion in the employed analyses.

## 5. Conclusions and Implications

The aim of the study was to gain knowledge about sociodemographic and health-related factors associated with long COVID. None of the sociodemographic variables were found to be associated with long COVID. Participants with long COVID experienced higher levels of psychological distress, fatigue, and perceived stress compared to participants without long COVID, and for two of the three health outcomes, the negative effects of long COVID were greater for men than for women. The results of the study have several implications. First, the study supports the notion that long-lasting COVID-19 symptoms are commonly experienced. Thus, long COVID may represent a growing challenge for health services, depending on vaccination rates and the ability of the vaccines to protect against the long-term impacts of COVID-19. Second, the long-term impacts of long COVID are by no means restricted to the most commonly experienced physical symptoms associated with acute COVID-19 infection, such as fever, shortness of breath, cough, sore throat, and loss of taste and smell. According to the results of this study, long COVID has a strong effect on levels of psychological distress, fatigue, and perceived stress. Third, related to psychological distress and fatigue, long COVID appears to have a stronger effect on men than on women. In view of men’s lower inclination to seek help, this may indicate that health services should pay extra attention to the potentially unnoticed needs for support among men experiencing long COVID. 

Possible lines of future research include further investigating the predictors of long COVID and investigating the trajectory of long COVID symptoms in longitudinal studies. Qualitative inquiries into the experience of people with long COVID may elicit more knowledge about the condition from an internal perspective. Future studies may investigate people’s perceptions and experiences of having the illness over an extended period of time. In the case of many common illnesses, people get sick and then get better. Long COVID breaks with this usual pattern, as people get sick and stay sick over time. The experience of not getting better from what has gradually become an endemic disease might lead to much frustration and ‘being sick and tired of being sick and tired’. Thus, the evolving psychological reactions to a long-term illness trajectory constitute an important area for future research on long COVID.

## Figures and Tables

**Figure 1 life-12-00901-f001:**
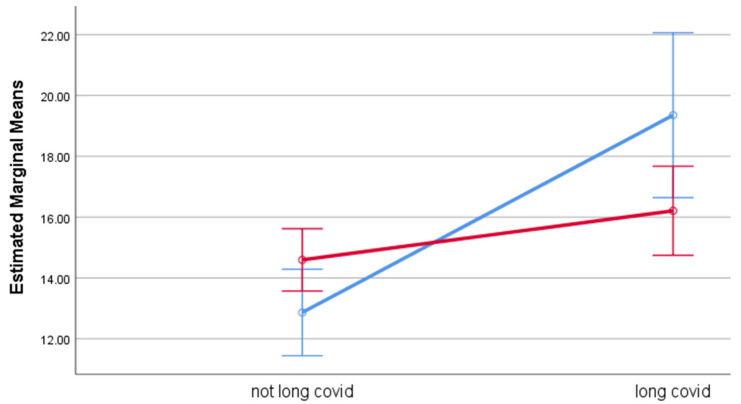
Estimated Marginal Means of General Health Questionnaire. Blue line represents males and red line represents females. Error bar is 95% CIs. Estimated means are adjusted by age.

**Figure 2 life-12-00901-f002:**
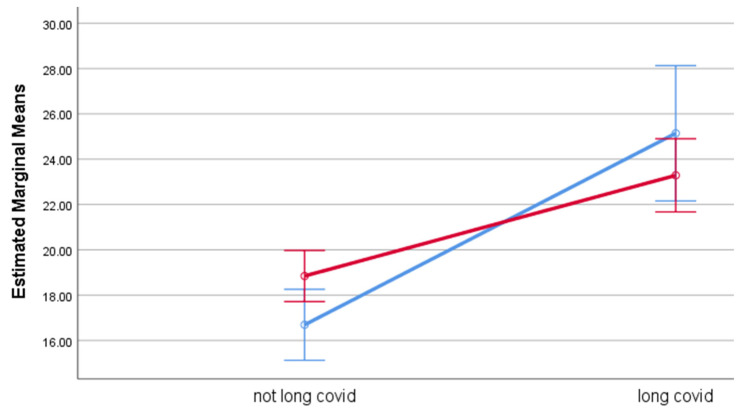
Estimated Marginal Means of the Chalder Fatigue Scale. Blue line represents males and red line represents females. Error bar is 95% CIs. Estimated means are adjusted by age.

**Figure 3 life-12-00901-f003:**
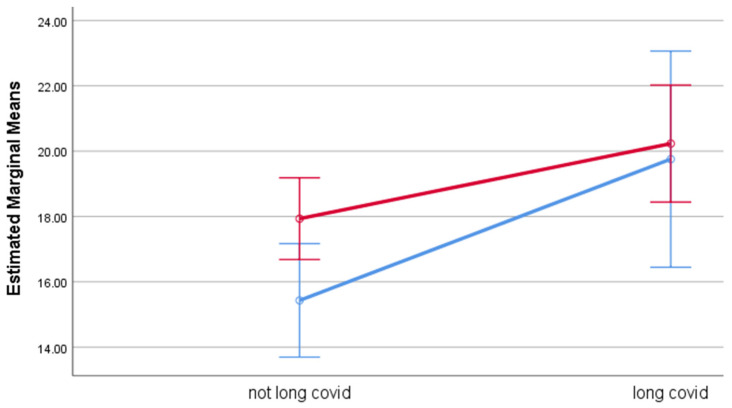
Estimated Marginal Means of the Perceived Stress Scale. Blue line represents males and red line represents females. Error bar is 95% CIs. Estimated means are adjusted by age.

**Table 1 life-12-00901-t001:** Number and proportions of participants with COVID-19 and long COVID by country.

Country	Participants (% ^a^)	COVID-19 Infection (% ^b^)	Long COVID (% ^c^)
Total sample	1649 (100.0)	310 (18.8)	87 (28.1)
Norway	242 (14.7)	13 (5.4)	7 (53.8)
UK	255 (15.5)	74 (29.0)	29 (40.3)
USA	915 (55.5)	220 (24.0)	49 (22.8)
Australia	237 (14.4)	3 (1.3)	2 (66.7)
*p * ^d^		<0.001	0.002

^a^ Percentage of all study participants. ^b^ Percentage of participants within country. ^c^ Percentage of participants within country who have had COVID-19 infection. ^d^ Chi-square tests. Significance values indicate the probability of proportions between countries being equal.

**Table 2 life-12-00901-t002:** Long COVID in sample subgroups (*n* = 303).

Subgroups	Long COVID *n* (%)	Not Long COVID *n* (%)	*p*
*Age group*			0.07
18–29 years	20 (37.7)	33 (62.3)	
30–39 years	18 (21.2)	67 (78.8)	
40–49 years	30 (29.1)	73 (70.9)	
50–59 years	16 (40.0)	24 (60.0)	
60–69 years	3 (18.8)	13 (81.3)	
70 years and over	0 (0.0)	6 (100.0)	
*Gender* ^a^			0.05
Male	19 (21.6)	69 (78.4)	
Female	65 (32.8)	133 (67.2)	
*Education level*			0.23
Lower education	21 (23.9)	67 (76.1)	
Higher education (bachelor’s degree or higher)	66 (30.7)	149 (69.3)	
*Spouse/partner*			0.28
No	30 (33.0)	61 (67.0)	
Yes	57 (26.9)	155 (73.1)	
*Employment*			0.51
No	22 (31.9)	47 (68.1)	
Yes	65 (27.8)	169 (72.2)	

Note: Chi-square tests. Significance values indicate the probability of proportions between groups being equal. ^a^
*n* = 286 due to some missing or non-binary responses to gender.

**Table 3 life-12-00901-t003:** Logistic regression analysis displaying adjusted associations between sociodemographic variables and long COVID (*n* = 286).

Independent Variables	Unadjusted			Adjusted		
	OR	95% CI	*p*	OR	95% CI	*p*
Higher age group	0.92	0.74–1.14	0.45	0.96	0.76–1.20	0.70
Female gender	1.78	0.99–3.20	0.06	1.75	0.97–3.17	0.07
Higher education	1.41	0.80–2.50	0.23	1.38	0.76–2.50	0.29
Spouse/partner	0.75	0.44–1.27	0.28	0.86	0.49–1.52	0.60
Employment	0.82	0.46–1.47	0.51	0.86	0.47–1.57	0.62
**Cox–Snell R^2^**					**0.019**	**0.35**
**Nagelkerke R^2^**					**0.028**	

**Table 4 life-12-00901-t004:** Effects of the independent variables on the three health outcomes (*n* = 286).

Independent Variables	Psychological Distress			Fatigue			Perceived Stress		
	*F* (*df*)	*p*	*ES*	*F* (*df*)	*p*	*ES*	*F* (*df*)	*p*	*ES*
Age group	4.53 (1)	<0.05	0.02	3.49 (1)	0.06	0.01	13.3 (1)	<0.001	0.05
Gender	0.62 (1)	0.43	0.00	0.02 (1)	0.88	0.00	1.83 (1)	0.18	0.01
Long COVID	20.2 (1)	<0.001	0.07	42.23 (1)	<0.001	0.13	9.06 (1)	<0.01	0.03
Long COVID × Gender	7.32 (1)	<0.01	0.03	4.08 (1)	<0.05	0.01	0.85 (1)	0.36	0.00
**R^2^ (Adjusted R^2^)**	**0.088 (0.075)**			**0.159 (0.147)**			**0.097 (0.084)**		

Note. ES is effect size, partial *η*^2^.

**Table 5 life-12-00901-t005:** Estimated marginal means for the three health outcomes by group (*n* = 286).

Independent Variables	Psychological Distress			
	*M*	95% CI	*p * ^a^	*ES * ^b^
COVID-19 status ^c^			<0.001	0.07
Not long COVID	13.7	12.9–14.6		
Long COVID	17.8	16.2–19.3		
Gender			0.43	0.00
Men	16.1	14.6–17.6		
Women	15.4	14.5–16.3		
Long COVID × gender			<0.01	0.03
Not long COVID men	12.9	11.4–14.3		
Long COVID men	19.4	16.6–22.1		
Not long COVID women	14.6	13.6–15.6		
Long COVID women	16.2	14.7–17.7		
	**Fatigue**			
	*M*	95% CI	*p*	*ES*
COVID-19 status ^c^			<0.001	0.13
Not long COVID	17.8	16.8–18.7		
Long COVID	24.2	22.5–25.9		
Gender			0.88	0.00
Men	20.9	19.2–22.6		
Women	21.1	20.1–22.1		
Long COVID × gender			<0.05	0.01
Not long COVID men	16.7	15.1–18.3		
Long COVID men	25.1	22.2–28.1		
Not long COVID women	18.8	17.7–20.0		
Long COVID women	23.3	21.7–24.9		
	**Perceived Stress**			
COVID-19 status	*M*	95% CI	*p*	*ES*
Not long COVID			<0.01	0.03
Long COVID	16.7	15.6–17.8		
Gender	20.0	18.1–21.9		
Men			0.18	0.01
Women	17.6	15.7–19.5		
Long COVID × gender	19.1	18.0–20.2		
Not long COVID men			0.36	0.00
Long COVID men	15.5	13.7–17.2		
Not long COVID women	19.8	16.4–23.1		
Long COVID women	17.9	16.7–19.2		
COVID-19 status ^c^	20.2	18.4–22.0		

^a^*p*-values indicate the probability of equality between groups. ^b^ ES is effect size; partial *η*^2^. ^c^ All results are adjusted by age.

## Data Availability

The data presented in this study will be made available on request from the corresponding author at the time of the research project’s completion. The data are not publicly available due to ongoing research and publications based on the data.

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
