# Peer review of "Self-Reported Long COVID in the General Population: Sociodemographic and Health Correlates in a Cross-National Sample"

_life, 2022, doi:10.3390/life12060901_

Round 1

Reviewer 1 Report

  1. The authors admit that, line 367, younger people were overrepresented, but to me, it was that older persons, especially over age 70, who were somehow missed in the survey sampling process.  The study offers little to say about long covid among the elderly.
  2. The authors consider a number of factors associated with long COVID and psychological symptoms.  I would suggest another factor - frustration with not getting well.  Normally, you get sick, you get better (if you don't die).  You get injured moderately and you get better with time.  Long COVID breaks that pattern - you get sick and stay sick.  Like being sick and tired of being sick and tired.  Some uncommon illnesses may act that way, but COVID has become endemic and common.  What is your common cold lasted for ten months?  People would get really irritated and discouraged by such a situation.

Author Response

We thank the reviewers for their guidance on this manuscript. All comments have been addressed below, and all changes in the revised manuscript have been performed using track changes. We look forward to hearing from you again.

*****************************************************************************

Reviewer 1 (R1): The authors admit that, line 367, younger people were overrepresented, but to me, it was that older persons, especially over age 70, who were somehow missed in the survey sampling process.  The study offers little to say about long covid among the elderly.

Authors’ response: We agree, and we have noted this point in the revised study limitations section.

R1: The authors consider a number of factors associated with long COVID and psychological symptoms.  I would suggest another factor - frustration with not getting well.  Normally, you get sick, you get better (if you don't die).  You get injured moderately and you get better with time.  Long COVID breaks that pattern - you get sick and stay sick.  Like being sick and tired of being sick and tired.  Some uncommon illnesses may act that way, but COVID has become endemic and common.  What is your common cold lasted for ten months?  People would get really irritated and discouraged by such a situation.

Authors’ response: Interesting point, which we have added to the revised Conclusion and Implications section.

Reviewer 2 Report

I accept the manuscript for its publication in this journal.

Author Response

We thank the reviewers for their guidance on this manuscript. All comments have been addressed below, and all changes in the revised manuscript have been performed using track changes. We look forward to hearing from you again.

**************************************************************************

Reviewer 2 (R2): I accept the manuscript for its publication in this journal.

Authors’ response: Thank you for the kind response.

Reviewer 3 Report

The reviewed paper deals with the COVID pandemic, especially with the “long COVID” phenomenon. People who have infected with COVID-19 sometimes report long lasting COVID symptoms like the loss of taste and smell. The authors of this paper examined whether sociodemographic as well as health-related differences between COVID-19 patients with and without long COVID could be observed. The authors did a cross-sectional online survey with a total sample of 1649 adults across four countries (Norway, UK, USA, and Australia). They analyzed the collected data descriptively and with logistic regression models and multivariate analyses of variances (MANOVAs). They found no sociodemographic differences between both groups (i.e., long COVID vs. no long COVID) but, however, observed differences on three health-related outcomes (psychological distress, fatigue, and perceived stress). COVID-patients with long COVID felt higher psychological distress, higer fatigue, and higer perceived stress than COVID-patients without long COVID. In addition, the authors introduced an interaction effect between ´long COVID (yes/no) and gener (men/women) and found that men reported significantly higher psychological distress and higher fatigue than women but not for perceived stress.

I really enjoyed reading this article and thank the authors for the opportunity to review their interesting paper. In my opinion, this paper deals with a very important and timely topic and is well structured. The introduction guides the reader smoothly into this topic, all necessary background information (i.e., citing appropriate literature) is given. It was a pleasure to read these sections!

However, I see several issues with the ending of this paper.I liked the ideas and the methodological setting (i.e., online survey) of this study and I think the reported data have potential. However, what I disliked was the interpretation and discussion of the empirical results. Frankly speaking, the authors are overselling their results. I found two major issues in the discussion of this paper:

(1)   The authors said in the statistic section that “statistical significance was set at p<0.05.” (page 5, line 212). However, later they wrote “The descriptively higher proportion of females with long COVID bordered statistical significance (p=0.05)” page 5, line 230) and “However, the different proportions of men and women with long COVID (22% and 33%, respectively) bordered towards statistical significance […]” (page 10, line 318). As a statistician, I felt very mad about this, because the results speak a different language, this is non-significant result. The authors wrote later in the conclusion section “[…] but a non-significant trend was found for the association between female gender and long COVID.” (page 11, line 389). This is a “no-go” in scientific research, the interpretation of “trends” resulted from non-significant results is strongly misleading. Honestly, there is no empirical evidence that there is an association between female gender and long COVID, there was no significant difference observed between men and women (p=0.05). That is the true story. Hence, I urge the authors not to mislead readers and to re-write these sections.

(2)   The next major issue targets the interpretation of interaction effects. The authors did three models with three different outcomes, namely psychological distress, fatigue, and perceived stress. Authors introduced an interaction effect long COVID x gender into the model in order to test their third hypothesis (“(iii) whether gender moderated the associations between long 119 COVID status and the health outcomes”, page 3, line 119). In the results section, only two (and not three!) interaction effects became significant (psychological distress: p<0.001, fatigue: p<0.05, and perceived stress: p=0.36, see Table 4, page 7, line 269). However, the authors wrote “While participants with long COVID generally perceived more psychological distress, fatigue, and stress than those without long COVID, differences were larger for men than for women” page 9, line 303), “[…] men’s perception of poorer health when having long COVID appears to include higher levels of psychological distress, fatigue and perceived stress” (page 11, line 363), “[…] and the negative effects of long COVID on the health outcomes were larger for men than for women” (page 11, line 392), and finally “Third, long COVID appears to have a stronger effect on men than on women” (page 11, line 401). Again, the authors are overselling their results. All these statements are not true, only for two out of three outcomes, namely psychological distress and fatigue, but not for perceived stress (see also Figure 3). Hence, I strongly urge the authors to re-write these sentences so that readers are not misled.

Besides these two major issues, I detected some smaller errors and miss-spellings (note, this list in not complete). Hence, I have some comments/suggestions that I hope will help the authors to further develop this line of work:

  • 1) Chapter 2 Materials and Methods: Something strange happened to the alignment of heading “2.2 Sample” (page 3, line 128). Please fix this.
  • 2) Table 1 (page 3, line 137): I detected an error in the third column (“COVID-19 infection”). The total sample is reported here as 303. However, the single numbers do not sum up to 303 but rather to 310 (13+74+220+3=310). I know, there were 7 missing values within the 310. However, this table is still wrong, it must be 310 and not 303. Please fix this. The percentage of 87 long COVID (28.7%) then refers to 303 (87/303=28.7%). Hence, I urge the authors to re-calculate percentages in Table 1. Otherwise, this is misleading the readership.
  • 3) Chapter 2.3 Measures: Sometimes the verbal scales are set in italic (e.g., “0=better than ususal”, page 4, line 173) and sometimes not (e.g., “4=very often”, page 5, line 187). Please use a consistent writing style throughout the whole manuscript.
  • 4) Table 3, Table 4, and Table 5: The authors use a very inconsistent style of bold. For example, see Table 4 (page 7, line 269): Column Fatgigue, why is ES not printed in bold? Please use a consistent writing style.
  • 5) Results (page 7, line 274): Sometimes fatigue (page 7, line 274) and sometimes with a capital F (Fatigue, page 7, line 283). Please use a consistent writing style throughout the whole manuscript.

Author Response

We thank the reviewers for their guidance on this manuscript. All comments have been addressed below, and all changes in the revised manuscript have been performed using track changes. We look forward to hearing from you again.

****************************************************************************

Reviewer 3 (R3): The reviewed paper deals with the COVID pandemic, especially with the “long COVID” phenomenon. People who have infected with COVID-19 sometimes report long lasting COVID symptoms like the loss of taste and smell. The authors of this paper examined whether sociodemographic as well as health-related differences between COVID-19 patients with and without long COVID could be observed. The authors did a cross-sectional online survey with a total sample of 1649 adults across four countries (Norway, UK, USA, and Australia). They analyzed the collected data descriptively and with logistic regression models and multivariate analyses of variances (MANOVAs). They found no sociodemographic differences between both groups (i.e., long COVID vs. no long COVID) but, however, observed differences on three health-related outcomes (psychological distress, fatigue, and perceived stress). COVID-patients with long COVID felt higher psychological distress, higer fatigue, and higer perceived stress than COVID-patients without long COVID. In addition, the authors introduced an interaction effect between ´long COVID (yes/no) and gener (men/women) and found that men reported significantly higher psychological distress and higher fatigue than women but not for perceived stress.

I really enjoyed reading this article and thank the authors for the opportunity to review their interesting paper. In my opinion, this paper deals with a very important and timely topic and is well structured. The introduction guides the reader smoothly into this topic, all necessary background information (i.e., citing appropriate literature) is given. It was a pleasure to read these sections!

Authors’ response: Thank you for the kind words.

R3: However, I see several issues with the ending of this paper. I liked the ideas and the methodological setting (i.e., online survey) of this study and I think the reported data have potential. However, what I disliked was the interpretation and discussion of the empirical results. Frankly speaking, the authors are overselling their results. I found two major issues in the discussion of this paper:

Authors response: Please see our comments to each of the listed issues below.

R3: The authors said in the statistic section that “statistical significance was set at p<0.05.” (page 5, line 212). However, later they wrote “The descriptively higher proportion of females with long COVID bordered statistical significance (p=0.05)” page 5, line 230) and “However, the different proportions of men and women with long COVID (22% and 33%, respectively) bordered towards statistical significance […]” (page 10, line 318). As a statistician, I felt very mad about this, because the results speak a different language, this is non-significant result.

Authors’ response: Thank you for pointing this out. The first mentioned statement has been removed altogether in the revised manuscript. The second statement has been modified in accordance with the reviewer’s view (see revised section 4.3).

R3: The authors wrote later in the conclusion section “[…] but a non-significant trend was found for the association between female gender and long COVID.” (page 11, line 389). This is a “no-go” in scientific research, the interpretation of “trends” resulted from non-significant results is strongly misleading. Honestly, there is no empirical evidence that there is an association between female gender and long COVID, there was no significant difference observed between men and women (p=0.05). That is the true story. Hence, I urge the authors not to mislead readers and to re-write these sections.

Authors’ response: Thank you for pointing this out. We have removed the relevant section in accordance with this guidance.

R3: The next major issue targets the interpretation of interaction effects. The authors did three models with three different outcomes, namely psychological distress, fatigue, and perceived stress. Authors introduced an interaction effect long COVID x gender into the model in order to test their third hypothesis (“(iii) whether gender moderated the associations between long COVID status and the health outcomes”, page 3, line 119). In the results section, only two (and not three!) interaction effects became significant (psychological distress: p<0.001, fatigue: p<0.05, and perceived stress: p=0.36, see Table 4, page 7, line 269). However, the authors wrote “While participants with long COVID generally perceived more psychological distress, fatigue, and stress than those without long COVID, differences were larger for men than for women” page 9, line 303),

Authors’ response: Thank you for pointing this out, we have revised the sentence (see section 4.1).

R3: “[…] men’s perception of poorer health when having long COVID appears to include higher levels of psychological distress, fatigue and perceived stress” (page 11, line 363),

Authors’ response: We have revised the sentence; see section 4.4.

R3: “[…] and the negative effects of long COVID on the health outcomes were larger for men than for women” (page 11, line 392)

Authors’ response: We have revised the sentence; see section 5.

R3: and finally “Third, long COVID appears to have a stronger effect on men than on women” (page 11, line 401). Again, the authors are overselling their results. All these statements are not true, only for two out of three outcomes, namely psychological distress and fatigue, but not for perceived stress (see also Figure 3). Hence, I strongly urge the authors to re-write these sentences so that readers are not misled.

Authors’ response: We have revised the sentence; see section 5.

R3: Besides these two major issues, I detected some smaller errors and miss-spellings (note, this list in not complete). Hence, I have some comments/suggestions that I hope will help the authors to further develop this line of work:

Authors’ response: Thank you for pointing out these mistakes, and we trust the editorial office will assist us further with any remaining details in the final stage of the process.

R3: Chapter 2 Materials and Methods: Something strange happened to the alignment of heading “2.2 Sample” (page 3, line 128). Please fix this.

Authors’ response: Alignment has been fixed.

R3: Table 1 (page 3, line 137): I detected an error in the third column (“COVID-19 infection”). The total sample is reported here as 303. However, the single numbers do not sum up to 303 but rather to 310 (13+74+220+3=310). I know, there were 7 missing values within the 310. However, this table is still wrong, it must be 310 and not 303. Please fix this. The percentage of 87 long COVID (28.7%) then refers to 303 (87/303=28.7%). Hence, I urge the authors to re-calculate percentages in Table 1. Otherwise, this is misleading the readership.

Authors’ response: Table 1 has been revised accordingly.

R3: Chapter 2.3 Measures: Sometimes the verbal scales are set in italic (e.g., “0=better than ususal”, page 4, line 173) and sometimes not (e.g., “4=very often”, page 5, line 187). Please use a consistent writing style throughout the whole manuscript.

Authors’ response: Italics have been removed, see revised section 2.3.

R3: Table 3, Table 4, and Table 5: The authors use a very inconsistent style of bold. For example, see Table 4 (page 7, line 269): Column Fatgigue, why is ES not printed in bold? Please use a consistent writing style.

Authors’ response: We are unsure how this problem occurred. We have fixed the problems in the revised tables.

R3: Results (page 7, line 274): Sometimes fatigue (page 7, line 274) and sometimes with a capital F (Fatigue, page 7, line 283). Please use a consistent writing style throughout the whole manuscript.

Authors’ response: Thank you for pointing out, we have addressed this issue throughout the revised manuscript.

Round 2

Reviewer 3 Report

I thank the authors for submitting a revised version of their manuscript entitled “Self-reported long COVID in the general population: sociodemographic and health correlates in a cross-national sample”.

The authors did a great job, improved their manuscript according to my suggestions, and responded to all my questions adequately. In my opinion, the revised manuscript increased a lot in comparison to the first version. Everything read more smoothly and I could follow authors’ argumentations easily, especially the non-significant gender effect and the interpretation of the interaction effects.

Before a final publication of this paper, I have two minor points left:

(1) Wrong column headings in tables: I detected that the authors replaced “GHQ” with “psychological distress” in the text (also “PSS” replaced with “perceived stress”). This is correct. Psychological distress is the latent construct which is measured with the instrument GHQ. Unfortunately, the authors missed to replace GHQ by psychological distress in Table 4 (page 7, line 280) and Table 5 (page 7, line 297; see column headings: not “GHQ”, “Fatigue”, and “Perceived stress” but rather “Psychological distress”, “Fatigue”, and “Perceived stress”). Please correct the two tables.

(2) Wrong use of abbreviations: The authors introduced the abbreviation “GHQ-12” for the 12-item General Health Questionnaire (see page 4, line 161), but wrote in the text only "GHQ" (e.g., page 5, line 207; page 6, line 256; page 7, line 274). Please replace every single “GHQ” in the text with the correct abbreviation “GHQ-12”. Please repeat this procedure for “PSS-10” (see page 4, line 185), i.e. replace “PSS” with “PSS-10” in the whole text.

Author Response

Reviewer 3 (R3): I thank the authors for submitting a revised version of their manuscript entitled “Self-reported long COVID in the general population: sociodemographic and health correlates in a cross-national sample”. The authors did a great job, improved their manuscript according to my suggestions, and responded to all my questions adequately. In my opinion, the revised manuscript increased a lot in comparison to the first version. Everything read more smoothly and I could follow authors’ argumentations easily, especially the non-significant gender effect and the interpretation of the interaction effects.

Authors’ response: Thank you for the kind – and quick – response!

R3: Before a final publication of this paper, I have two minor points left: Wrong column headings in tables: I detected that the authors replaced “GHQ” with “psychological distress” in the text (also “PSS” replaced with “perceived stress”). This is correct. Psychological distress is the latent construct which is measured with the instrument GHQ. Unfortunately, the authors missed to replace GHQ by psychological distress in Table 4 (page 7, line 280) and Table 5 (page 7, line 297; see column headings: not “GHQ”, “Fatigue”, and “Perceived stress” but rather “Psychological distress”, “Fatigue”, and “Perceived stress”). Please correct the two tables.

Authors’ response: Thank you for noticing. We have corrected the two tables according to this guidance.

R3: Wrong use of abbreviations: The authors introduced the abbreviation “GHQ-12” for the 12-item General Health Questionnaire (see page 4, line 161), but wrote in the text only "GHQ" (e.g., page 5, line 207; page 6, line 256; page 7, line 274). Please replace every single “GHQ” in the text with the correct abbreviation “GHQ-12”. Please repeat this procedure for “PSS-10” (see page 4, line 185), i.e. replace “PSS” with “PSS-10” in the whole text.

Authors’ response: Thank you for noticing. We have corrected the manuscript according to this guidance.

This manuscript is a resubmission of an earlier submission. The following is a list of the peer review reports and author responses from that submission.

Round 1

Reviewer 1 Report

It is a very important topic and is likely to be of great interest to readers. However, the definition of Long COVID, which is key to the analysis, was not explained to the participants and was registered exclusively according to their own interpretation. The lack of results from the various analyses could be due to this. Whether the main purpose of the study was to compare the health status of those who self-reported Long COVID and those who registered, or whether it was an attempt to find the causes of Long COVID, the objectives are also inconsistent and the description of the results gives the impression that the results are not logical. It may have been possible to obtain interesting results if the focus was on the health status of those who subjectively felt they were Long COVID.

Reviewer 2 Report

The only error I detected was at line 209 where it should say "in the USA" to keep consistent with line 210 where it says "in the UK".  Otherwise a nearly perfect paper. 

Reviewer 3 Report

The authors investigated the proportion of long COVID among 1649 adults in a cross-sectional online survey disseminated 25 openly in Norway, UK, USA, and Australia, and explored its correlation with sociodemographic factors and health outcomes. Though the sample size is relatively small for an online survey, this study still provides some useful information about long COVID based on multinational observation.